# Synthesis of Isomeric 3-Benzazecines Decorated with Endocyclic Allene Moiety and Exocyclic Conjugated Double Bond and Evaluation of Their Anticholinesterase Activity

**DOI:** 10.3390/molecules27196276

**Published:** 2022-09-23

**Authors:** Alexander A. Titov, Rosa Purgatorio, Arina Y. Obydennik, Anna V. Listratova, Tatiana N. Borisova, Modesto de Candia, Marco Catto, Cosimo D. Altomare, Alexey V. Varlamov, Leonid G. Voskressensky

**Affiliations:** 1Organic Chemistry Department, Peoples’ Friendship University of Russia (RUDN University), 6 Miklukho-Maklaya St, Moscow 117198, Russia; 2Department of Pharmacy-Pharmaceutical Sciences, University of Bari Aldo Moro, Via E. Orabona 4, 70125 Bari, Italy

**Keywords:** acetylcholinesterase, butyrylcholinesterase, monoamine oxidase A and B screening, anti-cholinesterase activity, azacyclic allenes, 3-benzazecines, -ylidene derivatives

## Abstract

Transformations of 1-methoxymethylethynyl substituted isoquinolines triggered by terminal alkynes in alcohols were studied and new 3-benzazecine-containing compounds synthesized, such as 6-methoxymethyl-3-benzazecines incorporating an endocyclic C6–C8 allene fragment and the -ylidene derivatives 6-methoxymethylene-3-benzazecines. The reaction mechanisms were investigated and a preliminary in vitro screening of their potential inhibitory activities against human acetyl- and butyrylcholinesterases (AChE and BChE) and monoamine oxidases A and B (MAO-A and MAO-B) showed that the allene compounds were more potent than the corresponding -ylidene ones as selective AChE inhibitors. Among the allenes, **3e** (R^3^ = CH_2_OMe) was found to be a competitive AChE inhibitor with a low micromolar inhibition constant value (*K*_i_ = 4.9 μM), equipotent with the corresponding 6-phenyl derivative **3n** (R^3^ = Ph, *K*_i_ = 4.5 μM), but 90-fold more water-soluble.

## 1. Introduction

Medium-sized nitrogen-containing heterocycles, i.e., 8-, 9-, 10-, 11-, and 12-membered rings are quite widespread in nature, since a number of alkaloids possess these core cyclic structures [1,2,3,4]. However, the chemical behavior of these heterocycles remains unclear, due to the fact that there are not enough effective methods for their synthesis [5,6,7,8,9] and the available ones are often limited to single examples, complexity of realization, or low group compatibility in substrates. Developing methods with broader applicability to the synthesis of such medium-sized heterocycles should helpfully support drug discovery and structure–activity relationship (SAR) studies. It is well known that the biological properties of compounds with 10-membered rings depend upon the conformation of the cycle [10], which in turn is mainly related to cumulated and conjugated bonds in molecular frameworks (Figure 1) [11,12] and by the presence of given pharmacophore features. The combination of these factors could open new opportunities for disclosing new medicinal hits targeted to biomolecules (e.g., enzymes, receptors), thus ultimately allowing the modification of the 3-benzazecine scaffold and possibly expanding their applicability in drug-discovery studies.

It should also be noted that heterocyclic nitrogen-containing allenes have not practically been studied. Moreover, while acyclic allenes are well known and successfully used in the syntheses of heterocycles, their cyclic analogues still require further detailed studies [13,14].

Previously, we have taken the first steps and succeeded in the construction of allene-containing 3-benzazecines [15]—a new type of allene A (R^3^ = Ph)—and later in our ongoing study observed some of their transformations [16,17]. It was shown that 8-alkyl(aralkyl)-substituted allene 3-benzazecines smoothly underwent transformation into 8-ylidene decorated derivatives in acetic acid (Figure 1). The purposes of this study were to synthesize new 3-benzazecine derivatives and investigate their chemical properties, as well as to preliminarily evaluate their in vitro biological properties as potential inhibitors of enzymes, which are drug targets related to neurological degenerative syndromes (e.g., Alzheimer and Parkinson diseases), namely, acetyl- and butyrylcholinesterases (AChE and BChE) and monoamine oxidases A and B (MAO-A and MAO-B).

## 2. Results and Discussion

Starting 1-methyl (isopropyl-, benzyl-, phenyl-, tolyl-, *p*-methoxyphenyl- and *p*-fluorophenyl)-1-methoxymethylethynyl-1,2,3,4-tetrahydroisoquinolines **2a–h** were obtained from 3,4-dihydroisoquinolinium methyl iodide **1** derived via the Bischler–Napieralski reaction [18], followed by alkylation and subsequent methoxymethyl ethynylation in the presence of cuprous bromide in methylene chloride (Figure 1, Table 1) [19].

We continued our study with estimating behavior of isoquinolines **2a**–**h** in reactions with terminal activated alkynes (methyl propiolate and acetylacetylene) in different solvents-trifluoroethanol, hexafluoroisopropanol, isopropanol, acetonitrile, or dichloromethane (Figure 2, Table 2).

In trifluoroethanol at 25 °C, isoquinolines **2a**, **2b**, **2g** with alkyl or benzyl substituents in the C-1 position reacted with methyl propiolate, readily forming benzazecines **3a**, **3b**, **3g** with an allene fragment as main products in 80–91% yield. However, reactions of isoquinolines **2c**–**f** with aryl substituent in the C-1 position under the same conditions did not proceed so clearly and led to the formation of mixtures of allene-containing benzazecines **3c**–**f** and 6-methoxymethylenebenzazecines **4c**–**f** in different ratios. The latter compounds were unexpected for us, as in previous work [16], we isolated only azecines with -ylidene fragment at C-8. We noticed that the prolongation in the reaction time led to the formation of the second product, compound **4**, so we tried to carry out the reactions quickly and immediately isolate target allene **3**.

Acetylacetylene also smoothly reacted with isoquinolines **2d**–**h** to provide allenes **3i**–**m** in moderate to high yields (Figure 2).

Previously, it was shown that 1-alkyl-1-phenylethynyltetrahydroisoquinolines under the action of methyl propiolate in hexafluoroisopropanol produced 8-ylidene-benzazecines [16], but in the case of 1-methoxymethylethynyl-substituted isoquinoline **2b**, the same reaction conditions led to the formation of benzazecine **3b** with an allene fragment in 40% yield. The reactions of isoquinolines **2b** and **2c** with alkynes in less acidic isopropanol proceeded slowly (4–10 days, 20 °C), resulting only in allenes **3b** and **3c** (Figure 2, Table 2). The formation of benzazecines with -ylidene moiety was not observed. The low yield of compound **3c** can be explained by a prolonged exposure of the reaction mixture in a proton solvent and, as a consequence, its strong tarring.

Acetonitrile and dichloromethane appeared not to be effective solvents for the transformations. Isoquinoline **2** did not react with methyl propiolate in either acetonitrile or dichloromethane. Reflux and MW irradiation could not solve the problem—the reactions in these solvents did not even start.

Based on the obtained experimental data, we presume that the reaction proceeds through the formation of zwitterion **I**, which exists in equilibrium with zwitterion **II** (Figure 3). The equilibrium position depends on the solvation ability of the solvent, substituents in the C-1 position of the isoquinoline, and delocalization of the anionic center.

In the case of acetylacetylene, the anionic center has greater nucleophilicity in comparison with one formed by methyl propiolate, so the reaction proceeds immediately after the formation of the initial ion **I**, leading to benzazecines **3i**–**m**.

In the case of methyl propiolate, delocalization of the anionic center promotes the formation of equilibrium and results in formation of a mixture of benzazecines **3c**–**f** and 6-ylidene decorated compounds **4c**–**f** (Figure 3).

The following step of the research was to study the behavior of obtained allene **3a** in acetic acid at 100 °C and microwave irradiation. It was of great interest to see whether the rearrangement in allene **3a** proceeds via a previously described route [17] or again prefers to yield 6-methoxymethylene benzazecines. In the abovementioned conditions, allene **3a** underwent rearrangement readily to give only 6-methoxymethylene benzazecine **4a** in 25% yield (Figure 4). The poor yield of the product can be explained by the use of more acidic protic solvent, such as AcOH, in which the intensive formation of tar products is observed. The short-term heating of reaction mixtures in an MW reactor does not improve the situation with the yields. We suggest that under the action of acetic acid, the allyl system is protonated, thus producing cation **III**, after stabilization of which 6-ylidene-substituted compound **4** is formed (Figure 4).

In previous work [12], the 10,11-dimethoxy derivative of the allene 3-benzazecine **3n** (scaffold **A**, R^3^ = Ph), bearing at C-8 the 4-methoxyphenyl group, was found to be the most potent competitive AChE-selective inhibitor (*K*_i_ about 4.5 μM). Herein, a number of newly and previously synthesized 3-benzazecine analogs, including either allene (Figure 1, scaffold **A**) or 6- and 8-ylidene (**B** and **C**) derivatives, were firstly assayed as inhibitors of AChE, BChE, and MAOs at 10 μM concentration. For compounds that attained at least 50% inhibition at 10 μM, IC_50_s were determined from the best-fitting inhibition-concentration curves (five scalar concentrations in the 0.1–50 μM range). The inhibition data only for the allene compounds, which achieved IC_50_ toward AChE in the low μM range, are reported in Table 3. Previously reported activities of **3n** and **3o** are also shown for comparison.

The only noteworthy activity was the AChE inhibition, for which the allene derivatives proved to be more potent than the -ylidene ones. The CO_2_Me esters **3d** and **3e** worked slightly better than the corresponding COMe ketones **3i** and **3j**. Compound **3e** bearing the polar methoxymethyl group at C-6 showed IC_50_ just double that of the corresponding 6-Ph analogue **3n**.

The Lineweaver–Burk plot of *h*AChE inhibition kinetics of the most active inhibitor **3e** showed a competitive mechanism (Figure 2), with inhibition constant *K*_i_ equal to 4.89 ± 0.47 μM, suggesting a preferential occupancy of the catalytic cavity of the enzyme by means of noncovalent interactions.

The enzymes’ inhibition assays showed that for all the tested compounds, the inhibitory effects toward both MAO isoforms, and BChE as well, were weak to nil in the low micromolar range. Possible antioxidant activities were also explored with the DPPH radical scavenging assay, where all compounds were inactive.

Interestingly, the replacement of the phenyl group at C-6 of **3n** with the more polar CH_2_OMe group in **3e**, while retaining the same inhibition potency, did improve the water solubility by 90 times. The experimental data (Table 4) showed a solubility in PBS at pH 7.4 for **3e** and **3n** equal to 17.4 and 0.2 μM, respectively. The hydrolytic stability of **3e** was quite good (half-life 4.5 h), though lower than the poorly soluble **3n** (half-life > 12 h).

The in silico prediction of ADME-related properties for **3e** and **3n** using the SwissADME tool [21] showed high gastrointestinal (GI) absorption, good permeation of the blood–brain barrier (BBB), and poor ability for compounds as P-glycoprotein 1 (P-gp) substrates. Indeed, tested in a P-gp assay, several similar analogues and **3n** itself proved to be potent inhibitors of P-gp in the nanomolar range. The two compounds were also predicted to inhibit cytochrome CYP3A4, a key liver enzyme responsible for oxidative detoxification of diverse xenobiotics, while no activity was suggested toward CYP2C19. Furthermore, the computational tool PAINS remover [22] did not alert for any PAINS (pan-assay interference compounds) for **3e** or **3n**.

## 3. Materials and Methods

### 3.1. Chemistry

#### 3.1.1. Materials and General Procedures

IR spectra were recorded on an Infralum FT-801 FTIR spectrometer in KBr tablets for crystalline compounds or in a film for amorphous compounds (ISP SB RAS, Novosibirsk, Russia). Elemental analysis was carried out on a Euro Vector EA-3000 elemental Analyzer (Eurovector, S.p.A., Milan, Italy) for C, H and N; experimental data agreed to within 0.04% of the theoretical values. ^1^H and ^13^C NMR spectra were acquired on a 600 MHz NMR spectrometer (JEOL Ltd., Tokyo, Japan) in CDCl_3_ for compounds with a solvent signal as internal standard (7.27 ppm for ^1^H nuclei, 77.2 ppm for ^13^C nuclei); peak positions were given in parts per million (ppm, *δ*). Mass spectra (LC-MS) of compounds were acquired on an Agilent 1100 LC/MSD VL system (electrospray ionization) (Agilent Technologies Inc., Santa Clara, CA, USA). Melting points were determined on an SMP-10 apparatus (Bibby Sterilin Ltd., Stone, UK) in open capillary tubes. Sorbfil PTH-AF-A-UF plates (Imid Ltd., Krasnodar, Russia) were used for TLC, visualization in an iodine chamber, or using KMnO_4_ and H_2_SO_4_ solutions. Silica gel (40–60 μm, 60 Å) Macherey-Nagel GmbH&Co (Loughborough, UK) was used for column chromatography. MW-assisted reactions were carried out in a Monowave 400 reactor from Anton Paar GmbH (Graz, Austria); the reaction temperature was monitored by an IR sensor; standard 10 mL G10 reaction vials, sealed with silicone septa, were used for the MW irradiation experiments. All reagents (Sigma-Aldrich, St. Louis, MO, USA; Merck, Darmstadt, Germany; J.T. Baker, Phillipsburg, NJ, USA), and fluorinated solvents (SIA P&M-Invest Ltd., Moscow, Russia) were used without additional purification.

#### 3.1.2. Synthesis of Benzazecines **3** and **4**

To compounds **2a**–**h** (1.7 mmol) was added 5 mL of 2,2,2-trifluoroethanol (hexafluoroisopropanol, isopropanol), then methyl propiolate or acetylacetylene (2.21 mmol) was added. In the case of methyl propiolate, the reaction proceeded at 25 °C and for acetylacetylene at 7 °C (Table 2). The reaction was carried out under argon atmosphere. The progress of the reaction was monitored by TLC (Sorbfil, 3:2 EtOAc–hexane). The solvent was removed under vacuum and residue was chromatographed on silica gel (1:5 EtOAc–hexane). Compounds **3a**–**m** and **4d**, **4f** were crystallized from Et_2_O.

*Methyl 3,8-dimethyl-10,11-dimethoxy-6-(methoxymethyl)-benzo[d]-3-aza-cyclodeca-4,6,7-triene-5-carboxylate* (**3a**): 0.507 g (80%); beige solid; mp 165–167 °C; *R*_f_ 0.60 (3:1, EtOAc–hexane); IR (KBr) ν 1961 (C=C=C), 1690 (C=O) cm^−1^; ^1^H NMR (CDCl_3_, 600 MHz) *δ* 7.41 (1H, s, H-4), 6.82 (1H, s, H Ar), 6.62 (1H, s, H Ar), 4.36–4.32 (1H, m, 2-CH_2_), 4.01 (1H, d, *J* = 11.6 Hz, C*H*_2_OCH_3_), 3.93 (1H, d, *J* = 11.6 Hz, C*H*_2_OCH_3_), 3.88 (3H, s, OCH_3_), 3.86 (3H, s, OCH_3_), 3.71 (3H, s, OCH_3_), 3.38–3.34 (1H, m, 2-CH_2_), 3.25 (3H, s, OCH_3_), 3.13 (3H, s, N-CH_3_), 2.89–2.83 (1H, m, 1-CH_2_), 2.75–2.69 (1H, m, 1-CH_2_), 2.10 (3H, s, CH_3_); ^13^C NMR (CDCl_3_, 150 MHz) *δ* 205.5, 170.0, 147.8, 147.6, 147.5, 131.1, 128.1, 113.1, 110.4, 97.7, 96.8, 94.2, 74.7, 58.7, 56.1, 56.0, 51.6, 51.2, 45.3, 31.4, 19.3; LCMS (ESI) *m/z* 374 [M + H]^+^; anal. C 67.61, H 7.19, N 3.81%, calcd for C_21_H_27_NO_5_, C 67.54, H 7.29, N 3.75%.

*Methyl 8-benzyl-3-methyl-10,11-dimethoxy-6-(methoxymethyl)-benzo[d]-3-aza-cyclodeca-4,6,7-triene-5-carboxylate* (**3b**): 0.694 g (91% from CF_3_CH_2_OH); white solid; mp 168–170 °C; *R*_f_ 0.55 (3:2, EtOAc–hexane); IR (KBr) ν 1955 (C=C=C), 1675 (C=O) cm^−1^; ^1^H NMR (CDCl_3_, 600 MHz) *δ* 7.41 (1H, s, H-4), 7.25–7.23 (4H, m, H Ph), 7.16 (1H, t, *J* = 7.1 Hz, H Ph), 6.81 (1H, s, H Ar), 6.58 (1H, s, H Ar), 4.39–3.35 (1H, m, 2-CH_2_), 4.00 (1H, d, *J* = 11.9 Hz, C*H*_2_OCH_3_), 3.98 (1H, d, *J* = 11.9 Hz, C*H*_2_OCH_3_), 3.84 (3H, s, OCH_3_), 3.78 (3H, s, OCH_3_), 3.76 (2H, s, C*H*_2_-Ph), 3.64 (3H, s, OCH_3_), 3.35–3.31 (1H, m, 2-CH_2_), 3.24 (3H, s, OCH_3_), 3.13 (3H, s, N-CH_3_), 2.88–2.82 (1H, m, 1-CH_2_), 2.70–2.64 (1H, m, 1-CH_2_); ^13^C NMR (CDCl_3_, 150 MHz) *δ* 206.4, 170.1, 147.7, 147.5, 139.6, 129.8, 129.0 (3C), 128.6, 128.3 (2C), 126.1, 112.9, 110.7, 101.8, 97.9, 94.1, 74.7, 58.8, 55.9 (2C), 51.6, 51.2, 45.3, 39.8, 31.3; LCMS (ESI) *m/z* 450 [M + H]^+^; anal. C 72.28, H 6.87, N 3.16%, calcd for C_27_H_31_NO_5_, C 72.14, H 6.95, N 3.12%.

*Methyl 3-methyl-8-phenyl-10,11-dimethoxy-6-(methoxymethyl)-benzo[d]-3-aza-cyclodeca-4,6,7-triene-5-carboxylate* (**3c**): 0.222 g (30% from CF_3_CH_2_OH); light yellow oil; *R*_f_ 0.53 (2:1, EtOAc–hexane); IR (KBr) ν 1943 (C=C=C), 1683 (C=O) cm^−1^; ^1^H NMR (CDCl_3_, 600 MHz) *δ* 7.42 (1H, s, H-4), 7.37 (2H, d, *J* = 8.1 Hz, H Ph), 7.31 (2H, t, *J* = 7.6 Hz, H Ph), 7.23 (1H, t, *J* = 7.6 Hz, H Ph), 6.75 (1H, s, H Ar), 6.70 (1H, s, H Ar), 4.44–4.41 (1H, m, 2-CH_2_), 4.21 (2H, s, C*H*_2_OCH_3_), 3.92 (3H, s, OCH_3_), 3.75 (3H, s, OCH_3_), 3.70 (3H, s, OCH_3_), 3.42–3.39 (1H, m, 2-CH_2_), 3.29 (3H, s, OCH_3_), 3.15 (3H, s, N-CH_3_), 2.94–2.89 (1H, m, 1-CH_2_), 2.85–2.80 (1H, m, 1-CH_2_); ^13^C NMR (CDCl_3_, 150 MHz) *δ* 207.1, 169.6, 147.9, 147.6, 147.5, 137.3, 129.6, 128.4 (2C), 128.0 (2C), 127.9, 126.9, 113.1, 112.5, 105.6, 100.5, 93.4, 74.5, 59.0, 56.0, 55.9, 51.5, 51.2, 45.1, 31.5; LCMS (ESI) *m/z* 436 [M + H]^+^; anal. C 71.55, H 6.89, N 3.14%, calcd for C_26_H_29_NO_5_, C 71.70, H 6.71, N 3.22%.

*Methyl 3-methyl-8-(4-methylphenyl)-10,11-dimethoxy-6-(methoxymethyl)-benzo[d]-3-aza-cyclodeca-4,6,7-triene-5-carboxylate* (**3d**): 0.359 g (47%); light yellow solid; mp 142–144 °C; *R*_f_ 0.53 (2:1, EtOAc–hexane); IR (KBr) ν 1935 (C=C=C), 1680 (C=O) cm^−1^; ^1^H NMR (CDCl_3_, 600 MHz) *δ* 7.42 (1H, s, H-4), 7.26 (2H, d, *J* = 8.1 Hz, H Ar), 7.13 (2H, d, *J* = 7.6 Hz, H Ar), 6.76 (1H, s, H Ar), 6.70 (1H, s, H Ar), 4.45–4.41 (1H, m, 2-CH_2_), 4.20 (2H, s, C*H*_2_OCH_3_), 3.91 (3H, s, OCH_3_), 3.75 (3H, s, OCH_3_), 3.70 (3H, s, OCH_3_), 3.42–3.39 (1H, m, 2-CH_2_), 3.29 (3H, s, OCH_3_), 3.14 (3H, s, N-CH_3_), 2.94–2.88 (1H, m, 1-CH_2_), 2.84–2.79 (1H, m, 1-CH_2_), 2.35 (3H, s, CH_3_); ^13^C NMR (CDCl_3_, 150 MHz) *δ* 206.8, 169.6, 147.8, 147.5, 147.4, 136.6, 134.2, 129.6, 129.0 (2C), 128.0, 127.9 (2C), 113.1, 112.4, 105.4, 100.3, 93.6, 74.6, 59.0, 56.0, 55.9, 51.5, 51.1, 45.0, 31.4, 21.1; LCMS (ESI) *m/z* 450 [M + H]^+^; anal. C 72.05, H 6.85, N 3.19%, calcd for C_27_H_31_NO_5_, C 72.14, H 6.95, N 3.12%.

*Methyl 3-methyl-10,11-dimethoxy-6-methoxymethyl-8-(4-methoxyphenyl)-benzo[d]-3-aza-cyclodeca-4,6,7-triene-5-carboxylate* (**3e**): 0.498 g (63%); orange oil; *R*_f_ 0.58 (1:2, EtOAc–hexane); IR (KBr) ν 1939 (C=C=C), 1682 (C=O) cm^−1^; ^1^H NMR (CDCl_3_, 600 MHz) *δ* 7.42 (1H, s, H-4), 7.28 (2H, d, *J* = 8.6 Hz, H Ar), 6.85 (2H, d, *J* = 8.6 Hz, H Ar), 6.75 (1H, s, H Ar), 6.69 (1H, s, H Ar), 4.44–4.40 (1H, m, 2-CH_2_), 4.19 (2H, s, C*H*_2_OCH_3_), 3.91 (3H, s, OCH_3_), 3.82 (3H, s, OCH_3_), 3.75 (3H, s, OCH_3_), 3.70 (3H, s, OCH_3_), 3.43–3.39 (1H, m, 2-CH_2_), 3.28 (3H, s, OCH_3_), 3.15 (3H, s, N-CH_3_), 2.93–2.88 (1H, m, 1-CH_2_), 2.84–2.79 (1H, m, 1-CH_2_); ^13^C NMR (CDCl_3_, 150 MHz) *δ* 206.5, 169.6, 158.7, 147.8, 147.5, 147.4, 129.5, 129.4, 129.1 (2C), 128.2, 113.8 (2C), 113.0, 112.4, 105.1, 100.3, 93.7, 74.6, 58.9, 56.0, 55.9, 55.3, 51.5, 51.1, 45.0, 31.4; LCMS (ESI) *m/z* 466 [M + H]^+^; anal. C 69.60, H 6.65, N 3.07%, calcd for C_27_H_31_NO_6_, C 69.66, H 6.71, N 3.01%.

*Methyl 3-methyl-10,11-dimethoxy-6-methoxymethyl-8-(4-fluorophenyl)-benzo[d]-3-aza-cyclodeca-4,6,7-triene-5-carboxylate* (**3f**): 0.185 g (24%); light yellow solid; mp 177–180 °C; *R*_f_ 0.39 (1:1, EtOAc–hexane); IR (KBr) ν 1941 (C=C=C), 1682 (C=O) cm^−1^; ^1^H NMR (CDCl_3_, 600 MHz) *δ* 7.42 (1H, s, H-4), 7.33–7.31 (2H, m, H Ar), 7.01–6.98 (2H, m, H Ar), 6.69 (2H, s, H Ar), 4.41–4.37 (1H, m, 2-CH_2_), 4.19 (1H, d, *J* = 11.9 Hz, C*H*_2_OCH_3_), 4.17 (1H, d, *J* = 11.9 Hz, C*H*_2_OCH_3_), 3.90 (3H, s, OCH_3_), 3.74 (3H, s, OCH_3_), 3.69 (3H, s, OCH_3_), 3.41–3.37 (1H, m, 2-CH_2_), 3.28 (3H, s, OCH_3_), 3.14 (3H, s, N-CH_3_), 2.95–2.88 (1H, m, 1-CH_2_), 2.84–2.79 (1H, m, 1-CH_2_); ^13^C NMR (CDCl_3_, 150 MHz) *δ* 206.8, 169.6, 162.9, 161.2, 148.1, 147.7 (2C), 133.4 (1C, d, *J* = 2.9 Hz), 129.6 (2C, d, *J* = 8.7 Hz), 127.9, 115.2 (2C, d, *J* = 20.2 Hz), 113.0, 112.6, 104.9, 100.8, 93.4, 74.5, 59.1, 56.1, 56.0, 51.5, 51.2, 45.2, 31.5; LCMS (ESI) *m/z* 454 [M + H]^+^; anal. C 68.80, H 6.32, N 3.15%, calcd for C_26_H_28_FNO_5_, C 68.86, H 6.22, N 3.09%.

*Methyl 3-methyl-8-isopropyl-6-(methoxymethyl)-benzo[d]-3-aza-cyclodeca-4,6,7-triene-5-carboxylate* (**3g**): 0.505 g (87%); yellow oil; *R*_f_ 0.75 (5:1, EtOAc–hexane); IR (KBr) ν 1951 (C=C=C), 1687 (C=O) cm^−1^; ^1^H NMR (CDCl_3_, 600 MHz) *δ* 7.42 (1H, s, H-4), 7.31 (1H, d, *J* = 8.1 Hz, H Ar), 7.24–7.21 (1H, m, H Ar), 7.16–7.13 (2H, m, H Ar), 4.44–4.39 (1H, m, 2-CH_2_), 4.04 (1H, d, *J* = 11.1 Hz, C*H*_2_OCH_3_), 3.95 (1H, d, *J* = 11.1 Hz, C*H*_2_OCH_3_), 3.72 (3H, s, OCH_3_), 3.36–3.32 (1H, m, 2-CH_2_), 3.21 (3H, s, OCH_3_), 3.13 (3H, s, N-CH_3_), 2.88–2.82 (1H, m, C*H*(CH_3_)_2_), 2.79–2.75 (2H, m, 1-CH_2_), 1.22 (3H, d, *J* = 6.9 Hz, CH_3_), 0.91 (3H, d, *J* = 6.9 Hz, CH_3_); ^13^C NMR (CDCl_3_, 150 MHz) *δ* 204.6, 170.2, 147.4, 138.0, 136.5, 130.0, 127.3, 126.9, 126.4, 109.3, 99.6, 94.6, 75.1, 58.9, 51.6, 51.1, 45.2, 31.8, 31.4, 22.2, 21.6; LCMS (ESI) *m/z* 342 [M + H]^+^; anal. C 73.76, H 8.11, N 4.19%, calcd for C_21_H_27_NO_3_, C 73.87, H 7.97, N 4.10%.

*Methyl 3-methyl-8-phenyl-6-(methoxymethyl)-benzo[d]-3-aza-cyclodeca-4,6,7-triene-5-carboxylate* (**3h**): 0.408 g (64%); beige solid; mp 148–150 °C; *R*_f_ 0.72 (5:1, EtOAc–hexane); IR (KBr) ν 1943 (C=C=C), 1668 (C=O) cm^−1^; ^1^H NMR (CDCl_3_, 600 MHz) *δ* 7.43 (1H, s, H-4), 7.36 (2H, d, *J* = 7.6 Hz, H Ar), 7.31 (2H, t, *J* = 7.6 Hz, H Ar), 7.27–7.25 (1H, m, H Ph), 7.24–7.22 (4H, m, H Ph), 4.45–4.41 (1H, m, 2-CH_2_), 4.22 (1H, d, *J* = 11.9 Hz, C*H*_2_OCH_3_), 4.20 (1H, d, *J* = 11.9 Hz, C*H*_2_OCH_3_), 3.71 (3H, s, OCH_3_), 3.47–3.43 (1H, m, 2-CH_2_), 3.28 (3H, s, OCH_3_), 3.15 (3H, s, N-CH_3_), 2.95–2.88 (2H, m, 1-CH_2_); ^13^C NMR (CDCl_3_, 150 MHz) *δ* 207.0, 169.6, 147.5, 137.2, 137.0, 136.0, 130.3, 129.8, 128.3 (2C), 128.1 (2C), 127.1, 126.9, 126.5, 105.5, 100.8, 93.1, 74.4, 59.0, 51.3, 51.1, 45.1, 31.8; LCMS (ESI) *m/z* 376 [M + H]^+^; anal. C 76.65, H 6.82, N 3.88%, calcd for C_24_H_25_NO_3_, C 76.77, H 6.71, N 3.73%.

*1-(3-Methyl-8-(4-methylphenyl)-10,11-dimethoxy-6-methoxymethyl-benzo[d]-3-aza-cyclodeca-4,6,7-trien-5-yl)ethanone* (**3i**): 0.368 g (50%); yellow solid; mp 156–159 °C; *R*_f_ 0.30 (EtOAc); IR (KBr) ν 1950 (C=C=C), 1641 (C=O) cm^−1^; ^1^H NMR (CDCl_3_, 600 MHz) *δ* 7.38 (1H, s, H-4), 7.26 (2H, d, *J* = 8.1 Hz, H Ar), 7.13 (2H, d, *J* = 8.1 Hz, H Ar), 6.73 (1H, s, H Ar), 6.69 (1H, s, H Ar), 4.43–4.39 (1H, m, 2-CH_2_), 4.13 (1H, d, *J* = 11.6 Hz, C*H*_2_OCH_3_), 4.11 (1H, d, *J* = 11.6 Hz, C*H*_2_OCH_3_), 3.90 (3H, s, OCH_3_), 3.73 (3H, s, OCH_3_), 3.42–3.38 (1H, m, 2-CH_2_), 3.29 (3H, s, OCH_3_), 3.18 (3H, s, N-CH_3_), 2.96–2.90 (1H, m, 1-CH_2_), 2.83–2.78 (1H, m, 1-CH_2_), 2.34 (3H, s, COCH_3_), 2.23 (3H, s, CH_3_); ^13^C NMR (CDCl_3_, 150 MHz) *δ* 206.1, 195.4, 148.0, 147.9, 147.5, 136.8, 134.1, 129.3, 129.1 (2C), 127.93, 127.90 (2C), 113.0, 112.5, 106.7, 105.7, 101.0, 75.0, 59.1, 55.9, 55.8, 51.6, 45.5, 31.2, 26.6, 21.1; LCMS (ESI) *m/z* 434 [M + H]^+^; anal. C 74.75, H 7.29, N 3.31%, calcd for C_27_H_31_NO_4_, C 74.80, H 7.21, N 3.23%.

*1-(3-Methyl-10,11-dimethoxy-6-methoxymethyl-8-(4-methoxyphenyl)-benzo[d]-3-aza-cyclodeca-4,6,7-trien-5-yl)ethanone* (**3j**): 0.557 g (73%); beige solid; mp 137–139 °C; *R*_f_ 0.26 (EtOAc); IR (KBr) ν 1938 (C=C=C), 1649 (C=O) cm^−1^; ^1^H NMR (CDCl_3_, 600 MHz) *δ* 7.38 (1H, s, H-4), 7.30 (2H, d, *J* = 9.1 Hz, H Ar), 6.86 (2H, d, *J* = 9.1 Hz, H Ar), 6.74 (1H, s, H Ar), 6.69 (1H, s, H Ar), 4.44–4.40 (1H, m, 2-CH_2_), 4.13 (1H, d, *J* = 11.9 Hz, C*H*_2_OCH_3_), 4.11 (1H, d, *J* = 11.9 Hz, C*H*_2_OCH_3_), 3.91 (3H, s, OCH_3_), 3.82 (3H, s, OCH_3_), 3.74 (3H, s, OCH_3_), 3.43–3.40 (1H, m, 2-CH_2_), 3.30 (3H, s, OCH_3_), 3.20 (3H, s, N-CH_3_), 2.96–2.90 (1H, m, 1-CH_2_), 2.84–2.78 (1H, m, 1-CH_2_), 2.24 (3H, s, COCH_3_); ^13^C NMR (CDCl_3_, 150 MHz) *δ* 206.0, 195.5, 158.9, 148.2, 148.0, 147.6, 129.5, 129.4, 129.3 (2C), 128.2, 114.0 (2C), 113.1, 112.7, 107.1, 105.6, 101.1, 75.2, 59.3, 56.1, 56.0, 55.4, 51.7, 45.7, 31.3, 26.7; LCMS (ESI) *m/z* 450 [M + H]^+^; anal. C 72.05, H 6.75, N 3.04%, calcd for C_27_H_31_NO_5_, C 72.14, H 6.95, N 3.12%.

*1-(3-Methyl-10,11-dimethoxy-6-methoxymethyl-8-(4-fluorophenyl)-benzo[d]-3-aza-cyclodeca-4,6,7-trien-5-yl)ethanone* (**3k**): 0.565 g (76%); light yellow solid; mp 164–166 °C; *R*_f_ 0.35 (EtOAc); IR (KBr) ν 1940 (C=C=C), 1650 (C=O) cm^−1^; ^1^H NMR (CDCl_3_, 600 MHz) *δ* 7.37–7.34 (3H, m, H-4, H Ar), 7.01 (2H, t, *J* = 8.6 Hz, H Ar), 6.70 (1H, s, H Ar), 6.69 (1H, s, H Ar), 4.42–4.38 (1H, m, 2-CH_2_), 4.13 (1H, d, *J* = 11.9 Hz, C*H*_2_OCH_3_), 4.10 (1H, d, *J* = 11.9 Hz, C*H*_2_OCH_3_), 3.92 (3H, s, OCH_3_), 3.74 (3H, s, OCH_3_), 3.42–3.38 (1H, m, 2-CH_2_), 3.30 (3H, s, OCH_3_), 3.20 (3H, s, N-CH_3_), 2.97–2.92 (1H, m, 1-CH_2_), 2.85–2.79 (1H, m, 1-CH_2_), 2.23 (3H, s, COCH_3_); ^13^C NMR (CDCl_3_, 150 MHz) *δ* 206.1, 195.3, 162.1 (1C, d, *J* = 247.1 Hz), 148.4, 148.2, 147.8, 133.3 (1C, d, *J* = 2.9 Hz), 129.7 (2C, d, *J* = 8.7 Hz), 129.4, 127.9, 115.4 (2C, d, *J* = 21.7 Hz), 113.0, 112.7, 106.9, 105.2, 101.5, 74.9, 59.3, 56.1, 56.0, 51.7, 45.7, 31.3, 26.5; LCMS (ESI) *m/z* 438 [M + H]^+^; anal. C 71.46, H 6.54, N 3.26%, calcd for C_26_H_28_FNO_4_, C 71.38, H 6.45, N 3.20%.

*1-(3-Methyl-6-methoxymethyl-8-isopropyl-benzo[d]-3-aza-cyclodeca-4,6,7-trien-5-yl)ethanone* (**3l**): 0.243 g (44%); colorless solid; mp 130–132 °C; *R*_f_ 0.45 (5:1, EtOAc–hexane). IR (KBr) ν 1941 (C=C=C), 1580 (C=O) cm^−1^; ^1^H NMR (CDCl_3_, 600 MHz) *δ* 7.45 (1H, s, H-4), 7.33 (1H, d, *J* = 7.6 Hz, H Ar), 7.25 (1H, td, *J* = 6.9, 1.7 Hz, H Ar), 7.16–7.13 (2H, m, H Ar), 4.39–4.35 (1H, m, 2-CH_2_), 3.96 (2H, s, C*H*_2_OCH_3_), 3.34–3.30 (1H, m, 2-CH_2_), 3.22 (3H, s, OCH_3_), 3.19 (3H, s, N-CH_3_), 2.90–2.86 (1H, m, C*H*(CH_3_)_2_), 2.85–2.83 (1H, m, 1-CH_2_), 2.82–2.77 (1H, m, 1-CH_2_), 2.29 (3H, s, COCH_3_), 1.27 (3H, d, *J* = 6.6 Hz, CH_3_), 0.93 (3H, d, *J* = 6.6 Hz, CH_3_); ^13^C NMR (CDCl_3_, 150 MHz) *δ* 204.1, 195.9, 147.9, 137.6, 136.1, 130.0, 127.2, 127.0 (2C), 126.5, 109.7, 100.6, 75.6, 59.1, 51.7, 45.6, 31.7, 31.4, 26.7, 22.3, 22.0; LCMS (ESI) *m/z* 326 [M + H]^+^; anal. C 77.61, H 8.25, N 4.25%, calcd for C_21_H_27_NO_2_, C 77.50, H 8.36, N 4.30%.

*1-(3-Methyl-6-methoxymethyl-8-phenyl-benzo[d]-3-aza-cyclodeca-4,6,7-trien-5-yl)ethanone* (**3m**): 0.305 g (50%); beige solid; mp 183–185 °C; *R*_f_ 0.47 (1:3, EtOAc–hexane); IR (KBr) ν 1942 (C=C=C), 1580 (C=O) cm^−1^; ^1^H NMR (CDCl_3_, 600 MHz) *δ* 7.39–7.37 (3H, m, H Ar and H-4), 7.32 (2H, t, *J* = 8.1 Hz, H Ar), 7.25–7.22 (5H, m, H Ph), 4.45–4.41 (1H, m, 2-CH_2_), 4.15 (1H, d, *J* = 11.9 Hz, C*H*_2_OCH_3_), 4.13 (1H, d, *J* = 11.9 Hz, C*H*_2_OCH_3_), 3.46–3.42 (1H, m, 2-CH_2_), 3.29 (3H, s, OCH_3_), 3.20 (3H, s, N-CH_3_), 2.98–2.88 (2H, m, 1-CH_2_), 2.24 (3H, s, COCH_3_); ^13^C NMR (CDCl_3_, 150 MHz) *δ* 206.4, 195.4, 148.1, 137.3, 136.9, 136.2, 130.4, 130.1, 128.5 (3C), 128.3 (2C), 127.3, 127.2, 126.8, 106.0, 101.6, 74.9, 59.3, 51.6, 45.7, 31.7, 26.7; LCMS (ESI) *m/z* 360 [M + H]^+^; anal. C 80.03, H 7.15, N 3.80%, calcd for C_24_H_25_NO_2_, C 80.19, H 7.01, N 3.90%.

*Methyl (4E,6E,7Z)-10,11-dimethoxy-6-(methoxymethylidene)-3-methyl-8-phenyl-1,2,3,6-tetrahydro-3-benzazecin-5-carboxylate* (**4c**): 0.237 g (32% from CF_3_CH_2_OH); yellow oil; *R*_f_ 0.52 (2:1, EtOAc–hexane); IR (KBr) ν 1685 (C=O) cm^−1^; ^1^H NMR (CDCl_3_, 600 MHz) *δ* 7.38 (1H, s, H-4), 7.28 (1H, s, H Ph), 7.24 (3H, t, *J* = 7.9 Hz, H Ph), 7.17 (1H, t, *J* = 7.1 Hz, H Ph), 6.64 (2H, br. s, H Ar and =C*H*-OCH_3_), 6.44 (1H, s, H Ar), 5.99 (1H, s, H-7), 4.15–4.08 (1H, m, 2-CH_2_), 3.91 (3H, s, OCH_3_), 3.75 (3H, s, OCH_3_), 3.74 (3H, s, OCH_3_), 3.50 (3H, s, OCH_3_), 2.96 (3H, s, N-CH_3_), 2.94–2.91 (1H, m, 2-CH_2_), 2.65–2.63 (1H, m, 1-CH_2_), 2.52–2.50 (1H, m, 1-CH_2_); ^13^C NMR (CDCl_3_, 150 MHz) *δ* 170.1, 152.9, 150.0, 148.0, 147.7, 142.5, 135.7, 134.6, 128.5, 128.1 (2C), 126.5, 126.2 (2C), 122.6, 113.8, 113.6, 112.0, 94.3, 60.3, 56.2 (2C), 55.8 (2C), 50.7, 32.3; LCMS (ESI) *m/z* 436 [M + H]^+^; anal. C 71.57, H 6.84, N 3.28%, calcd for C_26_H_29_NO_5_, C 71.70, H 6.71, N 3.22%.

Methyl (4E,6E,7Z)-3-methyl-10,11-dimethoxy-6-(methoxymethylidene)-8-(4-methylphenyl)-1,2,3,6-tetrahydro-3-benzazecin-5-carboxylate (**4d**): 0.267 g (35%); light yellow solid; mp 153–155 °C; R_f_ 0.52 (2:1, EtOAc–hexane); IR (KBr) ν 1680 (C=O) cm^−1^; ^1^H NMR (CDCl_3_, 600 MHz) δ 7.34 (1H, s, H-4), 7.17 (2H, d, J = 8.1 Hz, H Ar), 7.05 (2H, d, J = 8.1 Hz, H Ar), 6.63 (2H, br. s, H Ar and =CH-OCH_3_), 6.43 (1H, s, H Ar), 5.96 (1H, s, H-7), 4.15–4.08 (1H, m, 2-CH_2_), 3.91 (3H, s, OCH_3_), 3.75 (3H, s, OCH_3_), 3.74 (3H, s, OCH_3_), 3.49 (3H, s, OCH_3_), 2.96 (3H, s, N-CH_3_), 2.93–2.90 (1H, m, 2-CH_2_), 2.64–2.62 (1H, m, 1-CH_2_), 2.51–2.49 (1H, m, 1-CH_2_), 2.31 (3H, s, CH_3_); ^13^C NMR (CDCl_3_, 150 MHz) δ 170.1, 152.5, 149.9, 148.0, 147.6, 139.6, 136.2, 135.5, 134.8, 128.8 (2C), 128.4, 126.1 (2C), 121.6, 113.7, 113.6, 112.0, 94.3, 60.2, 56.2, 55.7 (2C), 50.6 (2C), 32.2, 21.0; LCMS (ESI) m/z 450 [M + H]^+^; anal. C 72.01, H 7.08, N 3.18%, calcd for C_27_H_31_NO_5_, C 72.14, H 6.95, N 3.12%.

Methyl (4E,6E,7Z)-10,11-dimethoxy-6-(methoxymethylidene)-8-(4-methoxyphenyl)-3-methyl-1,2,3,6-tetrahydro-3-benzazecin-5-carboxylate (**4e**): 0.221 g (28%); orange oil; R_f_ 0.53 (2:1, EtOAc–hexane); IR (KBr) ν 1679 (C=O) cm^−1^; ^1^H NMR (CDCl_3_, 600 MHz) δ 7.27 (1H, s, H-4), 7.19 (2H, d, J = 8.6 Hz, H Ar), 6.78 (2H, d, J = 8.6 Hz, H Ar), 6.63 (2H, br. s, H Ar and =CH-OCH_3_), 6.43 (1H, s, H Ar), 5.95 (1H, s, H-7), 3.91 (3H, s, OCH_3_), 3.78 (3H, s, OCH_3_), 3.74 (6H, s, OCH_3_), 3.72–3.70 (1H, m, 2-CH_2_); 3.49 (3H, s, OCH_3_), 2.96 (3H, s, N-CH_3_), 2.93–2.90 (1H, m, 2-CH_2_), 2.65–2.62 (1H, m, 1-CH_2_), 2.51–2.47 (1H, m, 1-CH_2_); ^13^C NMR (CDCl_3_, 150 MHz) δ 170.1, 158.6, 152.3, 149.9, 148.0, 147.7, 147.6, 135.3, 134.9, 129.2, 128.4, 127.3 (2C), 120.8, 113.6, 113.5 (2C), 112.1, 94.4, 60.2, 56.2, 55.8 (2C), 55.3 (2C), 50.6, 32.2; LCMS (ESI) m/z 466 [M + H]^+^; anal. C 69.55, H 6.85, N 3.23%, calcd for C_27_H_31_NO_6_, C 69.66, H 6.71, N 3.01%.

Methyl (4E,6E,7Z)-3-methyl-10,11-dimethoxy-6-(methoxymethylidene)-8-(4-fluorophenyl)-1,2,3,6-tetrahydro-3-benzazecin-5-carboxylate (**4f**): 0.362 g (47%); light yellow solid; mp 177–179 °C; R_f_ 0.38 (1:1, EtOAc–hexane); IR (KBr) ν 1682 (C=O) cm^−1^; ^1^H NMR (CDCl_3_, 600 MHz) δ 7.30 (1H, s, H-4), 7.23–7.21 (2H, m, H Ar), 6.94–6.91 (2H, m, H Ar), 6.64 (2H, br. s, H Ar and =CH-OCH_3_), 6.41 (1H, s, H Ar), 5.98 (1H, s, H-7), 4.16–4.11 (1H, m, 2-CH_2_), 3.91 (3H, s, OCH_3_), 3.75 (3H, s, OCH_3_), 3.74 (3H, s, OCH_3_), 3.49 (3H, s, OCH_3_), 2.96 (3H, s, N-CH_3_), 2.93–2.91 (1H, m, 2-CH_2_), 2.62–2.60 (1H, m, 1-CH_2_), 2.52–2.50 (1H, m, 1-CH_2_); ^13^C NMR (CDCl_3_, 150 MHz) δ 170.1, 162.8, 161.1, 152.9, 150.0, 148.2, 147.9, 138.8, 134.7, 134.5, 127.7 (2C, d, J = 7.2 Hz), 122.3, 114.9 (2C, d, J = 21.7 Hz), 113.7 (2C), 112.0, 94.3, 60.4, 56.3, 55.9 (2C), 50.7 (2C), 32.3; LCMS (ESI) m/z 454 [M + H]^+^; anal. C 68.75, H 6.17, N 3.15%, calcd for C_26_H_28_FNO_5_, C 68.86, H 6.22, N 3.09%.

#### 3.1.3. Transformation of Allene **3a** into 6-Methoxymethylidenebenzazecin **4a**

A solution of allene **3a** (0.4 mmol) in glacial acetic acid was placed into microwave reactor. The reaction was carried out for 20 min at 100 °C. The progress of the reaction was monitored by TLC (Sorbfil, 3:2 EtOAc-hexane). The solvent was removed under vacuum and the residue chromatographed on silica gel (1:5 EtOAc-hexane).

*Methyl (4E,6E,7Z)- 10,11-dimethoxy-6-(methoxymethylidene)-3,8-dimethyl-1,2,3,6-tetrahydro-3-benzazecin-5-carboxylate* (**4a**): 0.037 g (25%); brown oil; *R*_f_ 0.52 (2:1, EtOAc–hexane); IR (KBr) ν 1683 (C=O) cm^−1^; ^1^H NMR (CDCl_3_, 600 MHz) *δ* 6.71 (2H, s, H-4 and =C*H*-OCH_3_), 6.59 (1H, s, H Ar), 6.40 (1H, s, H Ar), 5.71 (1H, s, H-7), 3.86 (3H, s, OCH_3_), 3.76 (3H, s, OCH_3_), 3.68 (3H, s, OCH_3_), 3.47 (3H, s, OCH_3_), 2.94 (3H, s, N-CH_3_), 2.87–2.82 (2H, m, 1-CH_2_, 2-CH_2_), 2.66–2.62 (2H, m, 1-CH_2_), 2.03 (3H, s, CH_3_); ^13^C NMR (CDCl_3_, 150 MHz) *δ* 170.1, 150.8, 149.3, 148.0, 147.1, 137.2, 134.0, 126.1, 122.1, 113.8, 111.8, 111.6, 94.3, 60.0, 56.2 (2C), 55.9 (2C), 50.6, 31.7, 28.0; LCMS (ESI) *m/z* 374 [M + H]^+^; anal. C 67.41, H 7.08, N 3.38%, calcd for C_21_H_27_NO_5_, C 67.54, H 7.29, N 3.75%.

### 3.2. Inhibition of Cholinesterases and Inhibition of Monoamine Oxidases

#### 3.2.1. Inhibition of Cholinesterases

Inhibition of human recombinant AChE (2770 U/mg) or BChE from human serum (50 U/mg) was determined as described [23] using the Ellman spectrophotometric method in a 96-well plate procedure. Briefly, test compounds were incubated in phosphate buffer pH 8.0 in the presence of the enzyme and 5,5′-dithiobis-(2-nitrobenzoic acid) (DTNB) as the chromophoric reagent. Incubation samples were made in 96-well, flat-bottomed transparent polystyrene plates (Greiner Bio-One, Kremsmünster, Austria), at 25 °C for 20 min, and read at 412 nm using an Infinite M1000 Pro plate reader (Tecan, Cernusco s.N., Italy). For inhibition kinetics, four concentrations of compound **3e** (ranging from 0 to 15 μM), and six concentrations of acetylthiocholine (from 33 to 200 μM) were used. Inhibition data and kinetics were obtained as means ± SD from 3 independent experiments, using GraphPad Prism (version 5.00 for Windows; GraphPad Software, San Diego, CA, USA).

#### 3.2.2. Inhibition of Monoamine Oxidases

Inhibition of human recombinant monoamine oxidases A (250 U/mg) and B (59 U/mg; microsomes from baculovirus infected insect cells; Sigma Aldrich) was determined as already described [24], measuring the fluorescence of 4-hydroxyquinoline produced by MAOs in the oxidative deamination of substrate kynuramine. Briefly, compounds were tested in coincubation with MAO and kynuramine in phosphate buffer 390 mOsm pH 7.4, at 37 °C for 30 min. Assays were performed in 96-well black polystyrene plates (Greiner) using the Infinite M1000 Pro plate reader (Tecan). Inhibition data were obtained as means ± SD using GraphPad Prism.

### 3.3. Solubility and Hydrolytic Stability of ***3e*** and ***3n***

#### 3.3.1. Aqueous Solubility Measurement and U-HPLC Analytical Condition

The determination of kinetic solubility in aqueous buffer solution (50 mM phosphate buffer, pH 7.4, 0.15 M KCl) at 37 °C by U-HPLC was obtained as described [25], using a stock solution 10 mM in DMSO of compound (**3e** and **3n**) solubilized in PBS (50 mM) to final concentration of 200 μM. Following shaking of the suspension in an orbital shaker at 250 rpm for 2 h, the solution was separated by centrifugation (2500 rpm, 3 min) and filtered. Equal volume of solution was transferred into 1:1 (v/v) mixture of DMSO/PBS. The concentration of compound was determined by U-HPLC and UV detector (255 nm) comparing the peak area of external standard solution. All data were means of 3 independent experiments (± SEM). Analytical condition: mobile phase: MeOH/Ammonium formate 10 mM pH 4.5 (72:28); column: Kinetex C18, 150 × 2.1 mm, 2.6 µm; flow: 0.3 mL/min; injection: 2 µL (**3e**) and 5 µL (**3n**). HPLC analyses were performed on an Agilent U-HPLC 1260 Infinity Quaternary LC system (Agilent Technologies, Milan, Italy) (Table 4).

#### 3.3.2. Hydrolytic Stability in Water-Buffered Solution and U-HPLC Analytical Condition

Hydrolytic stability of compounds **3e** and **3n** was determined as described [26], using 10 mM stock solution in MeOH, solubilized in MeOH and aqueous buffer solution (50 mM phosphate buffer, pH 7.4 in 0.15 M KCl) to 25 µM final concentration, and incubated with shaking at 25 ± 0.5 °C. At appropriate time intervals, samples were withdrawn and analyzed by U-HPLC using a 1290 Infinity Quaternary LC system (Agilent Technologies, Milan, Italy) equipped with autosampler and photodiode array detector. A Phenomenex Kinetex C18 column 2.6 µm (150 × 2.1 mm i.d.) was used as stationary phase. The analyte was eluted with 8 min in isocratic mobile phase: MeOH/ammonium formate (10 mM, pH 4.5)/(68:32, *v*/*v*) at constant flow rate of 0.3 mL/min, injection volume: 2 µL (**3e**) and 5 µL (**3n**), UV detector: 255 nm. Pseudo-first-order rate constants (kobs) for the hydrolysis of the compound were calculated from the slopes of the linear plots of log (% remaining compound) against time. Each kinetic experiment was performed in triplicate (Table 4).

## 4. Conclusions

The conversion of 1-methoxymethylethynyl-substituted isoquinolines under the action of terminal alkynes in various alcohols was studied. It was shown that under the same reaction conditions, the transformations of the allene fragment depends on the substituent at C6 position in 3-benzazecines. A decrease in the yield of 6-methoxymethyl decorated allenes was observed in long-term and/or high-temperature reactions in protic solvents. A protocol for the synthesis of new 6-methoxymethyl substituted 3-benzazecines with an allene fragment and 6-methoxymethylene-3-benzazecines was developed.

A preliminary in vitro evaluation of the inhibition activity against the main target enzymes related to neurodegeneration revealed that the allene 3-benzazecine derivative **3e**, bearing the 6-methoxymethyl polar group, competitively inhibits AChE with a single-digit micromolar *K*_i_. Compound **3e** resulted in an inhibitor equipotent with the 6-phenyl analogue **3n**, but 90-fold more soluble in buffered aqueous solution at pH 7.4. This higher water-solubility property, joined with the potential of the core structure to inhibit P-gp efflux pumps and consequently to favor brain disposition [20], makes us confident that **3e** can be a candidate for further optimization of novel brain-permeant AChE inhibitors.

## Data Availability

All data presented in this study are available in the article and Appendix A.

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
