# Peer review of "Synthesis of Isomeric 3-Benzazecines Decorated with Endocyclic Allene Moiety and Exocyclic Conjugated Double Bond and Evaluation of Their Anticholinesterase Activity"

_molecules, 2022, doi:10.3390/molecules27196276_

Round 1
Reviewer 1 Report
This manuscript describes the synthesis of a series of 6-methoxymethyl-3-benzazecines via the transformation of ethynylisoquinolines with methyl propiolate and acetylacetylene. It was found that CF3CH2OH is the solvent of choice for this reaction. For methyl propiolate, the unexpected rearrangement was observed leading to 6-methoxymethylenebenzazecines. The proposed reaction mechanism clearly explains the formation of the products. Besides, invitro evaluation of inhibitory activities of compounds 3 against human acetyl- and butyrylcholinesterases (AChE and BChE) and monoamine oxidases A and B (MAO-A 21 and MAO-B) was carried out. Products 3 demonstrated IC50 toward AChE in the low μM range.
The structure of compounds was confirmed with the use of NMR (1H and 13C) spectra and HRMS data.
This work can be accepted in Molecules after minor revision.
1) How was the configuration of the exocyclic double bond of 4a established?
2) Why did trifluoroethanol promote the reaction?
3) Is it possible to explain the mechanism of biological activity of the obtained compounds toward AChE?
Author Response
Thanks to the reviewer for his/her comments and suggestions, which allowed us to improve the manuscript. A point-by-point reply is provided below.
1) How was the configuration of the exocyclic double bond of 4a established?
A: In the NMR spectrum signals of the H-7 proton and the H-6 methoxymethylidene group appear as singlets. The interaction constants between these protons could be observed in the Z-isomer. Also, of the two conformers, the E-isomer should be more advantageous for steric reasons. In the two-dimensional NOESY spectrum, there are no cross-peaks from protons of the H-7 and H-6 methoxymethylidene group.
2) Why did trifluoroethanol promote the reaction?
A: In our previous works it was shown that this reaction can proceed in any alcohol. But in trifluoroethanol we obtained the target product with a quantitative yield, and the reaction took place in less time. We suggest that, due to the high acidity, trifluoroethanol stabilizes the negative charge of the initial zwitterion I, which facilitates the Aza-Claisen reaction. Confirmation of this can be found in the literature.
3) Is it possible to explain the mechanism of biological activity of the obtained compounds toward AChE?
A: Compound 3e displayed competitive inhibition on hAChE (Figure 2). This means that it occupies the catalytic cavity of the enzyme, competing with the substrate, by means of non-covalent interactions with amino acids surrounding the catalytic site. A short sentence has been added to the text.
Reviewer 2 Report
The manuscript molecules-1901928 describes the transformation of 1-methoxymethylethynyl substituted isoquinolines into 3-benzazecine-containing compounds. Reaction mechanisms were investigated and in vitro studies for the evaluation of the inhibitory activity of the compounds on AChE, BChE and MAO-A and MAO-B were conducted, together with in silico calculations of the ADME-related properties for the two most active compounds.
The manuscript is well written and well organized. The experimental methods are described with enough detail to allow repetition of the experiments, and conclusions are well-supported by the experimental results. Detailed structural characterization of the obtained compounds is presented.
Only some minor aspects should be improved before publication.
a) For a better comprehension, the last sentence in the abstract should be rewritten as “Among the allenes, 3e (R3 = CH2OMe) was found as competitive AChE inhibitor with a low micromolar inhibition constant value (Ki = 4.9 μM), equipotent with the corresponding 6-phenyl derivative 3n (R3 = Ph, Ki = 4.5 μM) but 90-fold more water-soluble.
b) In the keywords please replace the abbreviations AChE, BChE, MAOs A and B by the corresponding keywords.
c) Page 2, line 55, please confirm if the term 8-alkyl(aralkyl) is correct.
d) Page 2, line 65, please check if the name of compound 1 is correct. I think it should be 2-methyl-3,4-dihydroisoquinolin-2-ium iodide or at least 3,4-dihydroisoquinolinium methyliodide, and delete the first “of” in the sentence of line 67.
e) Page 3, line 84 change 8-C by C-8 to agree with the style used in previous lines (C-1, lines 78 and 80). There are other examples that need to be changed along he manuscript to keep the same style, example line 111, pg. 4.
f) Table 2, lines 1, 2 and 13, I suppose the correct temperature is 7 ºC and not 25 ºC, according to the paragraph on lines 77-79. Space between the number and the unit is required when indicating the temperature. Please change.
g) Page 3, lines 95-98, the authors stated that reaction of 2b in hexafluoroisopropanol led to the formation of 6-methoxymethylenebenzazecine 4b in 40% yield. This is not in agreement with line 3 of Table 2. Please check if any correction is needed.
h) Scheme 4. The caption of the scheme should be on the same page as the scheme. Please check the reaction time for the reaction in microwave. In scheme 4 the reaction time is 5 min. but in the experimental procedure (page 11, line 417) it is said that the reaction time is 20 min.
i) Page 6. Lines 181 and 182, please indicate the abbreviation for high gastrointestinal absorption (GI) and define BBB before the abbreviation.
j) Page 7, Experimental procedure lines 214-215, the reaction was not always performed at 7 ºC. Maybe consider the other temperatures or make a reference to table 2 for other solvents and temperatures.
Once this minor corrections have been done, the manuscript can be accepted for publication.
Author Response
Thanks to the referee for the constructive comments and suggestions. A point-by-point reply is provided below.
a) For a better comprehension, the last sentence in the abstract should be rewritten as “Among the allenes, 3e (R3 = CH2OMe) was found as competitive AChE inhibitor with a low micromolar inhibition constant value (Ki = 4.9 μM), equipotent with the corresponding 6-phenyl derivative 3n (R3 = Ph, Ki = 4.5 μM) but 90-fold more water-soluble.”
A: Text changed.
b) In the keywords, please replace the abbreviations AChE, BChE, MAOs A and B by the corresponding keywords.
A: Changes done.
c) Page 2, line 55, please confirm if the term 8-alkyl(aralkyl) is correct.
A: The use of this term is correct, since in order to form 8-ylidenes, the presence of a benzyl substituent in the C-8 position with various radicals in the benzene ring is necessary.
d) Page 2, line 65, please check if the name of compound 1 is correct. I think it should be 2-methyl-3,4-dihydroisoquinolin-2-ium iodide or at least 3,4-dihydroisoquinolinium methyliodide and delete the first “of” in the sentence of line 67.
A: Necessary changes done.
e) Page 3, line 84 change 8-C by C-8 to agree with the style used in previous lines (C-1, lines 78 and 80). There are other examples that need to be changed along he manuscript to keep the same style, example line 111, pg. 4.
A: Changes done.
f) Table 2, lines 1, 2 and 13, I suppose the correct temperature is 7 ºC and not 25 ºC, according to the paragraph on lines 77-79. Space between the number and the unit is required when indicating the temperature. Please change.
A: The reaction proceeds at 25 °C with methyl propiolate; the necessary changes have been added to the text.
g) Page 3, lines 95-98, the authors stated that reaction of 2b in hexafluoroisopropanol led to the formation of 6-methoxymethylenebenzazecine 4b in 40% yield. This is not in agreement with line 3 of Table 2. Please check if any correction is needed.
A: The formation of allene 3b occurs under these conditions; the necessary changes have been added to the text.
h) Scheme 4. The caption of the scheme should be on the same page as the scheme. Please check the reaction time for the reaction in microwave. In scheme 4 the reaction time is 5 min. but in the experimental procedure (page 11, line 417) it is said that the reaction time is 20 min.
A: The reaction proceeds for 20 minutes; the necessary changes have been added to the text.
i) Page 6. Lines 181 and 182, please indicate the abbreviation for high gastrointestinal absorption (GI) and define BBB before the abbreviation.
A: Change done.
j) Page 7, Experimental procedure lines 214-215, the reaction was not always performed at 7 ºC. Maybe consider the other temperatures or make a reference to Table 2 for other solvents and temperatures.
A: Changes done.